# Mapping the Anti-Cancer Activity of α-Connexin Carboxyl-Terminal (aCT1) Peptide in Resistant HER2+ Breast Cancer

**DOI:** 10.3390/cancers16020423

**Published:** 2024-01-19

**Authors:** Kimberly M. Baker, Melissa Abt, Emma H. Doud, Adrian L. Oblak, Elizabeth S. Yeh

**Affiliations:** 1Department of Biology, University of Indianapolis, Indianapolis, IN 46227, USA; bakerkm@uindy.edu; 2Department of Pharmacology and Toxicology, Indiana University School of Medicine, Indianapolis, IN 46202, USA; 3Melvin and Bren Simon Comprehensive Cancer Center, Indiana University School of Medicine, Indianapolis, IN 46202, USA; edoud@iu.edu; 4Department of Biochemistry and Molecular Biology, Indiana University School of Medicine, Indianapolis, IN 46202, USA; 5Department of Radiology and Imaging Sciences, Indiana University School of Medicine, Indianapolis, IN 46202, USA; aoblak@iupui.edu; 6Stark Neurosciences Research Institute, Indiana University School of Medicine, Indianapolis, IN 46202, USA

**Keywords:** connexin 43, breast cancer, HER2, gap junction

## Abstract

**Simple Summary:**

Breast cancer is a leading cause of death for women worldwide. This study aims to examine the mechanisms for the anti-cancer effects of a therapeutic peptide agent targeted to connexin 43 (Cx43) called alpha-connexin carboxyl-terminal peptide (aCT1). Findings from this study confirm that aCT1 promotes gap junction intercellular communication and Cx43 protein stabilization and that the peptide binds to Zonal Occludens-1 (ZO-1), consistent with prior reports on the mechanism of action of this agent. New findings from this study identify novel aCT1-interacting proteins and the effect of aCT1 on breast cancer signaling.

**Abstract:**

Connexin 43 (Cx43) is a protein encoded by the *GJA1* gene and is a component of cell membrane structures called gap junctions, which facilitate intercellular communication. Prior evidence indicates that elevated *GJA1* expression in the HER2-positive (HER2+) subtype of breast cancer is associated with poor prognosis. Prior evidence also suggests that HER2+ breast cancers that have become refractory to HER2-targeted agents have a loss of Cx43 gap junction intercellular communication (GJIC). In this study, a Cx43-targeted agent called alpha-connexin carboxyl-terminal peptide (aCT1) is examined to determine whether GJIC can be rescued in refractory HER2+ breast cancer cells. A proposed mechanism of action for aCT1 is binding to the tight junction protein Zonal Occludens-1 (ZO-1). However, the true scope of activity for aCT1 has not been explored. In this study, mass spectrometry proteomic analysis is used to determine the breadth of aCT1-interacting proteins. The NanoString nCounter Breast Cancer 360 panel is also used to examine the effect of aCT1 on cancer signaling in HER2+ breast cancer cells. Findings from this study show a dynamic range of binding partners for aCT1, many of which regulate gene expression and RNA biology. nCounter analysis shows that a number of pathways are significantly impacted by aCT1, including upregulation of apoptotic factors, leading to the prediction and demonstration that aCT1 can boost the cell death effects of cisplatin and lapatinib in HER2+ breast cancer cells that have become resistant to HER2-targeted agents.

## 1. Introduction

Gap junctions are membrane channels composed of six connexin proteins. The role of gap junctions is to connect adjoining cells allowing for intercellular communication through the exchange of ions, second messengers, and small metabolites (generally < 1 kDa in size). Connexins are transmembrane proteins that contain extracellular and cytoplasmic tails and loops. Each cell connected by a gap junction contains a six-protein channel with a pair of these channels composing the gap junction. The human genome encodes for 21 connexin genes.

Connexin 43 (Cx43) is one of the most well-studied connexin proteins and has been studied in a variety of human tissues and diseases [1,2,3,4,5,6,7,8,9]. We and others have studied the role of Cx43 in breast cancer [10,11,12,13,14,15,16,17,18,19,20,21,22,23,24,25,26]. Prior studies suggest that, in tumors, Cx43 expression can be either decreased or increased and that Cx43’s localization can shift intracellularly from the cell membrane [19,22,27,28,29,30,31,32]. Maintaining gap junction intercellular communication (GJIC) by keeping Cx43 within gap junction plaques in breast cancer cells has been suggested to elicit a “bystander” effect [33,34,35,36]. Under this paradigm, promoting GJIC would result in an amplification of therapeutic responses [37,38]. However, the development and evaluation of Cx43-targeted agents in breast and other cancers is complicated because Cx43 appears to have both canonical gap junction roles as well as non-canonical intracellular roles [4,39]. Furthermore, findings suggest that gap junctions could play a role in the establishment of metastatic colonies in tissues where tumor cells have disseminated [23,40].

There are a number of connexin-based therapeutics in various phases of development for the treatment of diseases including breast cancer [41,42,43]. α-connexin carboxyl-terminal (aCT1) is a therapeutic peptide modeled on the sequence of the C-terminal tail of Cx43. Application of aCT1 was shown to increase gap junction size and GJIC [12,44]. Prior work shows the anti-cancer effects of aCT1 in the brain, melanoma, and breast cancer [12,45,46]. These reports suggest that aCT1 may have anti-cancer effects of its own and is quite effective at boosting the impact of other anti-cancer agents when added in combination, consistent with the concept of the bystander effect. We and others have made prior observations that the upregulation of Cx43 in HER2-positive (HER2+) breast cancer correlates with poorer outcomes [9,21]. Our previous findings have also shown that refractory HER2+ breast cancer cells downregulate Cx43-mediated GJIC [21]. In this study, we examine the mechanisms by which aCT1 exerts its anti-cancer properties by conducting proteomic and genomic analyses of resistant HER2+ breast cancer cells treated with aCT1. 

## 2. Materials and Methods

### 2.1. Chemicals and Reagents

Biotinylated α-connexin carboxyl-terminal (aCT1) peptide was purchased from AmbioPharm Inc. (North Augusta, SC, USA) and was reconstituted in 1X PBS (Corning, Corning, NY, USA), as previously described [12]. The aCT1 peptide corresponds to a short sequence at the Cx43 C-terminus linked to an antennapedia internalization sequence. Lapatinib was purchased from LC Laboratories (Woburn, MA, USA) and reconstituted in DMSO. Cisplatin, purchased from Selleck Chemicals LLC (Houston, TX, USA), was reconstituted in distilled PBS. All reagents were aliquoted and stored at −80 °C. Cycloheximide was purchased from Acros Organic (Thermo Scientific Chemicals, Waltham, MA, USA). Antibodies: Cx43 (C6219 and MAB3067, Millipore Sigma, St. Louis, MO, USA); β-tubulin (sc-55529, Santa Cruz Biotechnology, Dallas, TX, USA); ZO-1 (8193, Cell Signaling Technology, Danvers, MA, USA).

### 2.2. Cell Culture

JIMT-1 cells obtained from AddexBio (San Diego, CA, USA) were cultured in Dulbecco’s modified Eagle’s medium (DMEM) (Corning, Corning, NY, USA) supplemented with 10% heat-inactivated fetal bovine serum (Gibco, Waltham, MA, USA), 1% L-glutamine, and 1% penicillin/streptomycin (Corning, Corning, NY, USA). Cells were maintained in a 37 °C humidified incubator with 5% CO_2_.

### 2.3. Parachute Assay

JIMT-1 acceptor cells were plated in 96-well plates at a density of 20,000 cells per well and additional JIMT-1 donor cells were plated in 10 cm plates. The next day, both the acceptor and donor cells were treated with either vehicle (PBS) or 250 µM aCT1. On the third day, the donor JIMT-1 cells were loaded with calcein AM (BD Biosciences, Bedford, MA, USA) or vehicle, treated in serum-free DMEM for 30 min, and then counted using a LUNA-FL dual fluorescence cell counter (Logos Biosystems, Annandale, VA, USA). Five thousand calcein AM-loaded donor cells or vehicle donor cells were added to the adherent acceptor cells in the 96-well plates and incubated for 6 h. Cells were then trypsinized and counted using a LUNA-FL cell counter. N = 3 independent experiments with at least 8 replicates per treatment.

### 2.4. Cell Death Assay

JIMT-1 cells were plated in 96-well plates at a density of 20,000 cells per well and treated the following day with 250 µM aCT1 or PBS for 24 h prior to analysis by trypan blue exclusion and cell counting using a LUNA-FL cell counter. N = 3 independent experiments with at least 8 replicates per treatment.

### 2.5. Proteomic Analysis

Total protein was isolated from JIMT-1 cells treated with 100 µM aCT1 or PBS for 4 h using lysis buffer containing 50 mM Tris pH7.5, 150 mM NaCl, 1 mM EDTA, and 15 Triton X-100. Lysate was cleared by centrifugation and then normalized to 10 mg per sample. Streptavidin matrix (Thermo Scientific, Waltham, MA, USA) was added to the lysate and samples were incubated at 4 °C overnight. Streptavidin–biotin–aCT1 pulldown was then washed in lysis buffer and PBS prior to proteomic analysis. 

#### 2.5.1. Sample Preparation

A total of 6 on-bead samples (n = 3 of CTL JIMT and n = 3 of ACT1 JIMT) were submitted to the IUSM Center for proteome analysis where proteins were denatured in 8 M urea and 100 mM Tris-HCl, pH 8.5, and reduced with 5 mM tris(2-carboxyethyl)phosphine hydrochloride (TCEPSigma-Aldrich, St. Louis, MO, USA, Cat No: C4706) for 30 min at room temperature. Samples were then alkylated with 10 mM chloroacetamide (CAA, Sigma Aldrich Cat No: C0267) for 30 min at room temperature in the dark, prior to dilution with 50 mM Tris-HCl, pH 8.5, to a final urea concentration of 2 M for trypsin/Lys-C-based overnight protein digestion at 37 °C (0.5 µg protease, mass spectrometry grade, Promega Corporation, Madison, WI, USA, Cat No: V5072). Digestions were acidified with trifluoroacetic acid (TFA, 0.5% *v*/*v*) and desalted on Pierce C18 spin columns (Thermo Fisher, Waltham, MA, USA, Cat No: 89870) with a wash of 0.5% TFA followed by elution in 70% acetonitrile 0.1% formic acid (FA).

#### 2.5.2. Nano-LC-MS/MS 

Mass spectrometry was performed utilizing an EASY-nLC 1200 HPLC system (SCR: 014993, Thermo Fisher Scientific) coupled to an Exploris 480™ mass spectrometer with a FAIMSpro interface (Thermo Fisher Scientific). One-fifth of each fraction was loaded onto a 25 cm IonOpticks-TS column (Ionopticks Aurora Ultimate TS 25 cm) at 350 nL/min. The gradient was held at 5% B for 5 min (mobile phases A: 0.1% formic acid (FA), water; B: 0.1% FA, 80% acetonitrile (Thermo Fisher Scientific Cat No: LS122500)), then increased from 4 to 30% B over 98 min, 30 to 80% B over 10 min, held at 80% for 2 min, and dropped from 80 to 4% B over the final 5 min. The mass spectrometer was operated in positive ion mode, default charge state of 2, advanced peak determination on, and lock mass of 445.12003. Three FAIMS CVs were utilized (−40 CV; −55 CV; −70 CV), each with a cycle time of 1.3 s and with identical MS and MS2 parameters. Precursor scans (*m*/*z* 375–1500) were performed with an orbitrap resolution of 120,000, RF lens% 40, automatic maximum inject time, standard AGC target, minimum MS2 intensity threshold of 5 × 10^3^, MIPS mode to peptide, including charges of 2 to 7 for fragmentation with 30 sec dynamic exclusion. MS2 scans were performed with a quadrupole isolation window of 1.6 *m*/*z*, 30% HCD CE, 15,000 resolution, standard AGC target, automatic maximum IT, and fixed first mass of 110 *m*/*z*. 

#### 2.5.3. Mass Spectrometry Data Analysis

Resulting RAW files were analyzed in Proteome Discover™ 2.5 (Thermo Fisher Scientific) with a Homo sapiens reference proteome FASTA (downloaded from Uniprot 13 May 2022, 78,806 entries) plus common contaminants (73 entries) [47]. Default Minora Feature selection settings were used. SEQUEST HT searches were conducted with a maximum number of 3 missed cleavages, a precursor mass tolerance of 10 ppm, and a fragment mass tolerance of 0.02 Da. Static modifications used for the search were carbamidomethylation on cysteine (C). Dynamic modifications included oxidation of methionine (M), and acetylation, methionine loss, or methionine loss plus acetylation on protein N-termini. Percolator False Discovery Rate was set to a strict peptide spectral match FDR setting of 0.01 and a relaxed setting of 0.05. In the consensus workflow, the default feature mapper settings of a maximum RT shift of 10 min and a minimum S/N threshold of 5 were used. Precursor ion quantification settings use unique razor peptides for quantification, base precurson ion abundance for intensity, and no normalization of scaling. For Quant rollup and hypothesis testing, summed abundances were used with pairwise ratio-based calculations and no imputation. The resulting abundance values for each sample, abundance ratio and log2(abundance ratio) values, and respective *p*-values from Proteome Discover™ were exported to Microsoft Excel (version 2308).

### 2.6. Gene Expression Analysis

Total RNA was isolated from JIMT-1 cells treated with 250 µM aCT1 or PBS for 24 h using the GeneJet RNA isolation kit (Thermo Fisher Scientific, Waltham, MA, USA). Gene expression was analyzed using the Breast Cancer 360™ Panel, composed of 758 genes relevant to breast cancer biology and 18 housekeeping genes, on the nCounter platform (NanoString Technologies, Seattle, WA, USA) as described by the manufacturer. NanoString nSolver 4.0 software was used for the analysis of gene expression values, which were normalized using the housekeeping genes and log2 transformed. The software incorporates the R statistics program. N = 3 control or aCT1-treated samples per group were analyzed.

### 2.7. Mammosphere Assay

JIMT-1 cells were plated in low-adhesion 96-well plates (BrandTech Scientific Inc., Essex, CT, USA) at a density of 500 cells per well in serum-free DMEM/Ham’s F-12 50/50 medium with L-glutamine (Corning, Corning, NY, USA) supplemented with 1× B27 (Gibco, Waltham, MA, USA), 20 ng/mL basic fibroblast growth factor (Gibco, Waltham, MA, USA), 20 ng/mL epidermal growth factor (Sigma-Aldrich, Inc., St. Louis, MO, USA), and 2 µg/mL heparin (StemCell Technologies Inc., Vancouver, BC, Canada). On the day of plating, cells were treated with either vehicle (PBS), lapatinib (100 nM), aCT1 (50 µM), cisplatin (500 nM), a combination of aCT1 and lapatinib, or a combination of aCT1 and cisplatin. After 14 days, the number of spheres with a diameter ≥50 μm was counted using a TCM 400 inverted light microscope (Labomed, Fremont, CA, USA) at 40× magnification. N ≥ 3 independent experiments with at least 8 replicates per treatment.

### 2.8. Statistical Analysis

GraphPad Prism software (version 8) was used for statistical analysis. Data were expressed as the mean ± standard error (SE) where n = number of independent experiments. Statistical differences among the groups were analyzed using either Student’s *t*-test for comparisons between two groups or one-way analysis of variance (ANOVA) followed by Tukey’s multiple comparisons test. A statistically significant difference was set at *p* < 0.05. (* *p* <0.05, ** *p* ≤ 0.01, *** *p* ≤0.001, **** *p* ≤ 0.0001, ns = not significant).

## 3. Results

### 3.1. aCT1 Promotes Gap Junction Intercellular Communication in Refractory HER2+ Breast Cancer Cells

Prior work determined that HER2+ breast cancer cells that have become resistant to HER2-targeted agents have reduced Cx43-mediated GJIC [21]. Therefore, we aimed to determine whether we could reestablish Cx43-mediated GJIC using the Cx43-targeted agent aCT1. To evaluate the effects of aCT1 in this context, we performed a cell-coupling assay (Figure 1A) using the JIMT-1 cell line, which is a HER2+ breast cancer cell line that was isolated from a patient that had developed resistance to trastuzumab (aka Herceptin) [48]. Cells were treated with PBS (CTL) or aCT1 and assessed for coupling. aCT1-treated cells showed increased coupling, suggesting that GJIC was restored (Figure 1B).

### 3.2. Reestablishment of Gap Junction Intercellular Communication by aCT1 Promotes Cx43 Protein Stabilization

A number of reports have indicated that the half-life of Cx43 is within a range of <4 h (hrs) [49,50,51,52,53,54,55]. However, the stability of Cx43 appears to be extended by being within the gap junction plaque structure [56]. These reported observations prompted us to investigate whether the increased GJIC we observed upon aCT1 treatment would lead to increased Cx43 protein stabilization. We treated JIMT-1 cells with PBS or aCT1 for up to 4 h in the presence of the protein synthesis inhibitor cycloheximide to look at protein degradation. We then evaluated the levels of Cx43 protein expression by Western blotting and saw that aCT1 treatment prevented Cx43 degradation, whereas Cx43 levels declined in the absence of aCT1 (Figure 2A). Cells treated with aCT1 and cycloheximide had higher levels of Cx43 than CTL cells treated with cycloheximide (Figure 2B).

### 3.3. Proteomic Analysis of Proteins that Bind to aCT1

The aCT1 sequence is modeled after the C-terminal sequence of Cx43, which is reported to interact with the tight junction protein ZO-1 [57,58,59]. The reported mechanism of action for aCT1 leading to increased GJIC is binding to ZO-1, thereby physically impairing ZO-1 from preventing gap junction accretion [44]. Additionally, the c-terminus of Cx43 is known to interact with a large number of proteins including src, caveolin, β-catenin, p120-catenin, and tubulin [8,60,61,62]. Therefore, we wanted to assess interacting partners of aCT1 specifically in JIMT-1, resistant HER2+ breast cancer cells. We treated cells with PBS or aCT1 for 4 h and 24 h. Since aCT1 is biotin-tagged, we then pulled down aCT1 using a streptavidin matrix. Coomassie staining to assess for proteins bound to aCT1 showed a large number of proteins present in the 4 h sample but not in the 24 h sample, which resembled the no-peptide control samples (Figure 3A). Although peptide therapeutics are quite promising (currently >80 FDA-approved peptide-based therapies), a major challenge is that peptides have poor stability [63,64]. Consistent with this concept and the reported half-life of Cx43, aCT1 was not detected in the 24 h samples.

To identify proteins bound to aCT1, we submitted aCT1 pulldown samples generated from JIMT-1 cells treated for 4 h for sequencing at the Indiana University School of Medicine Center for Proteome Analysis. To confirm the validity of our assay, we assessed the aCT1 pulldown samples for binding to ZO-1 and confirmed that aCT1 binds ZO-1 by Western blot (Figure 3B). Overall, the proteomic analysis identified >2500 proteins, and 354 of those proteins were bound to aCT1 with high confidence (*p*-value < 0.0001; Appendix A). We also confirmed the presence of ZO-1 bound to aCT1 in our proteomics data. Caveolin, β-catenin, p120-catenin, and tubulin were also among the >2500 proteins identified.

Because the mechanism of action for aCT1 is that it binds to ZO-1 and other proteins that would normally bind to the C-terminus of Cx43, we assessed the potential consequences of the aCT1 binding partners by performing pathway analysis using the STRING protein–protein interaction network tool (string-db.org; accessed on 26 October 2023) to examine the proteins significantly bound to aCT1 compared to CTL [8]. Here, we report the top pathways identified (Table 1). The full analysis can be found in Appendix A. The results from Gene Ontology (GO) Biological Process, GO Biological Function, KEGG, and Reactome analysis indicated that proteins bound to aCT1 are involved in gene expression, RNA binding, ribosome, and RNA metabolism, respectively (Table 1). Gene Ontology (GO) Biological Process analysis suggests that the proteins we identified that bound to aCT1 are involved in gene expression. This finding is consistent with prior reports that implicate Cx43 as having gene expression regulatory functions [4,65,66]. In particular, Cx43 was found to directly interact with a basic transcription factor 3 (BTF3) to directly regulate transcription of the gene that encodes for N-cadherin, CDH2 [66]. aCT1 bound to BTF3 in our analysis, consistent with these prior findings (Appendix A).

### 3.4. aCT1 Impacts Gene Expression

To further assess the ability of aCT1 to alter gene expression, we performed an analysis to determine whether breast cancer signaling pathways are altered in response to aCT1 treatment. Here, we assessed JIMT-1 cells treated with PBS or aCT1 using the NanoString nCounter Breast Cancer 360 panel. Although our data suggest that aCT1 stability is limited beyond 4 h, we were interested to observe if any longer-term effects could be seen in response to peptide treatment. Therefore, we collected JIMT-1 cells after 24 h of aCT1 treatment and analyzed the RNA using the nCounter technology. Using a *p*-value cutoff of *p* < 0.05 as significant, we found that 52 of the ~700 genes represented in the panel were significantly altered 24 h after aCT1 treatment (Appendix A and Figure 4A). nCounter analysis includes 23 different breast cancer-related pathways with specific genes that have been curated to represent those pathways. Directed differential expression analysis comparing aCT1 vs. a baseline of non-aCT1-treated CTL cells showed that aCT1-treated cells had upregulated genes in seventeen of those pathways and downregulated genes in six (Figure 4B). We also analyzed the 52 altered genes using STRING and found that the results from GO Biological Process, GO Biological Function, KEGG, and Reactome analysis indicated that aCT1-altered genes were involved in the cell cycle, protein binding, cancer pathways, and cell junction organization, respectively (Table 2 and Appendix A). Of note, the top two upregulated pathways in the nCounter analysis from aCT1-treated samples are epigenetic regulation and apoptosis, and the top pathway that showed decreased expression was adhesion and migration (Figure 4B).

Since N-cadherin, which is regulated by the Cx43 binding protein BTF3, regulates adhesion and migration, we looked for the expression levels of CDH2, which encodes for N-cadherin within our gene expression data. We found that CDH2 is significantly reduced in aCT1-treated samples (Figure 4C). 

To further investigate these findings and the anti-cancer effects of aCT1, we evaluated apoptotic genes that were upregulated by aCT1. We observed that the pro-apoptotic factors BAX and BBC3 (aka PUMA) were significantly upregulated by aCT1 (Figure 4C). Prior reports indicate that the reestablishment of GJIC in resistant HER2+ breast cancer cells will resensitize these cells to HER2 inhibitors [21]. Similarly, the upregulation of PUMA promotes the sensitization of cancers, including ovarian and melanoma, to cisplatin [67,68,69]. Therefore, we assessed the ability of aCT1 alone and in combination with lapatinib and cisplatin to impair the tumorigenic properties of the JIMT-1 cells using measurements of cell death and tumorsphere formation. First, we assessed whether aCT1 alone could effectively induce cell death and found that cell death was increased by aCT1 (Figure 5A). We then performed a mammosphere assay as a surrogate for non-adherent tumor cell growth and treated cells with aCT1, lapatinib, or cisplatin alone, or aCT1 in combination with lapatinib or cisplatin. We found that the combination of aCT1 with either agent was significantly more effective at impairing mammosphere growth than any agent alone (Figure 5B).

## 4. Discussion

Numerous studies have evaluated Cx43 in breast cancer or other human cancers but fewer have looked at targeting Cx43 with a targeted agent [39,41]. Only a handful of prior studies evaluated the agent aCT1 for use in cancer [12,46,70]. Prior reports have indicated that the mechanism of action for aCT1 is the binding of ZO-1, leading to gap junction accretion, which promotes GJIC [44]. The predicted consequence of aCT1 application leading to increased GJIC is Cx43 stabilization [56]. In this study, we show for the first time in a resistant HER2+ breast cancer cell line that has lost GJIC that aCT1 reestablished GJIC, leading to Cx43 stabilization. Using proteomic and traditional protein–protein biochemical binding analyses, we also demonstrate that aCT1 directly binds ZO-1. New findings from this study show that aCT1 binds to a wide range of proteins, some of which have been previously identified as Cx43 C-terminal interaction proteins as predicted, but also factors that have potentially not been previously studied in such a capacity. We have highlighted previously reported binding partners of the C-terminus of Cx43 in this study: src, caveolin, β-catenin, p120-catenin, and tubulin. These were identified in our screen. We also noted that there was a report that cyclin E1 and p110β are also binding partners of either the C-terminus of Cx43 or aCT1 [45,71]. We did not identify these factors in our proteomic analysis. However, our genomic analysis showed decreased PI3K signaling, which is consistent with the effect aCT1 would have by binding to p110β. 

Canonical signaling places Cx43 at the cell membrane in gap junctions and hemichannels. However, non-canonical functions of Cx43 in relation to its interactome are also reported [8,72,73,74,75]. A prior study in prostate cancer showed that Cx43 elevation was associated with increased metastatic potential of prostate cancer cells. However, application of aCT1 did not have anti-cancer effects and it was suggested by the authors that there may be gap junction-independent (i.e., non-canonical) roles for Cx43 in prostate cancer [76]. Our studies found that aCT1, like the C-terminus of Cx43, binds to a host of proteins, and as expected, a portion of these interacting proteins are pertinent to Cx43 functions at the membrane, while others are important for other cellular functions, most notably gene expression and RNA regulation. Others have reported a direct ability of Cx43 to enter the nucleus and regulate transcription [66,77]. The example that we presented was the regulation of BTF3. Cx43 binds to BTF3 to regulate the transcription of CDH2, which encodes for the N-cadherin protein [66]. aCT1 bound to BTF3 in our proteomic analysis and consistent with this, the gene that encodes N-cadherin, CDH2, was significantly reduced in aCT1-treated samples in our gene expression analysis using the NanoString nCounter Breast Cancer 360 panel. This finding is consistent with the role of Cx43 in promoting N-cadherin expression as a pro-cancer function and with the predicted effect of aCT1 having an anti-cancer effect. Our data show that aCT1 binds to BTF3, presumably impairing its ability to interact with Cx43. As a consequence, we would expect CDH2 expression to also be reduced in aCT1-treated samples, and this is what we observed.

An area of ongoing investigation is Cx43′s role in metastasis. Some findings suggest that gap junctions could play a role in the establishment of metastatic colonies in tissues where tumor cells have disseminated [23,40]. In support of this conjecture, prior studies with aCT1 in cells, rodents, and humans indicate that aCT1 has wound-healing properties [78,79,80,81,82]. However, other studies suggest an anti-angiogenic effect of aCT1 [83] and a pro-metastatic effect of preventing Cx43 GJIC [21].

Like other studies that have reported the anti-cancer effects of aCT1 [45,84], our prior studies have implicated Cx43 as a therapeutic target in HER2+ breast cancer [12,21]. Based on our prior findings, it is predicted that the reestablishment of GJIC in resistant HER2+ breast cancer cells, which are gap junction-deficient, will resensitize cells to HER2-targeted agents. We now show this prediction to be true using a mammophere assay, which acts as a type of in vitro surrogate for tumor growth. Our findings demonstrate that the combination of aCT1 and lapatinib is highly effective at preventing sphere development and growth compared to CTL or single agents on their own. This finding is also consistent with previous studies that report an additive or synergistic effect of aCT1 with other anti-cancer chemotherapeutic agents [12,45,84]. Furthermore, new gene expression data indicates that aCT1 induces the expression of the pro-apoptotic genes BAX and BBC3. This finding is consistent with prior reports and the suggested outcome of BBC3 induction leading to the sensitization of cancer cells to cisplatin [67,68,69], we find that aCT1 combined with cisplatin has a greater effect than either single agent on its own. Taken together, our findings provide a more complete picture of the scope of anti-cancer effects induced by the Cx43-targeted agent, aCT1, painting a picture of how to move forward with implementing a strategy for the use of this agent for HER2+ breast cancers, and perhaps other human cancers. 

## 5. Conclusions

In this study, we examined the mechanisms for the anti-cancer effects of a therapeutic peptide agent targeted to Cx43 called aCT1. Findings from this study confirm that aCT1 promotes gap junction intercellular communication and Cx43 protein stabilization and binds to ZO-1 in resistant HER2+ breast cancer cells. These findings are consistent with prior reports on the mechanism of action of this agent. New findings from this study identify novel aCT1-interacting proteins and the effects of aCT1 on breast cancer signaling. nCounter analysis using the NanoString Breast Cancer 360 panel showed that aCT1 significantly altered apoptosis, leading to improved efficacy of lapatinib and cisplatin in combination with aCT1. 

## Figures and Tables

**Figure 1 cancers-16-00423-f001:**
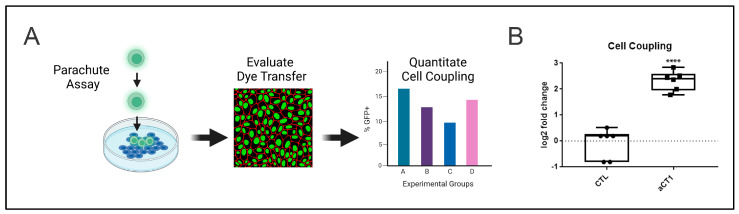
aCT1 promotes gap junction intercellular communication in refractory HER2+ breast cancer cells. (**A**) Parachute assay used to determine cell coupling as a measure of gap junction intercellular communication. Created with Biorender. (**B**) Quantitation of parachute assay from JIMT-1 cells treated with PBS as a control (CTL) or 250 μM aCT1 for 6 hrs. Student’s *t*-test, unpaired with Welch’s correction **** *p* < 0.0001.

**Figure 2 cancers-16-00423-f002:**
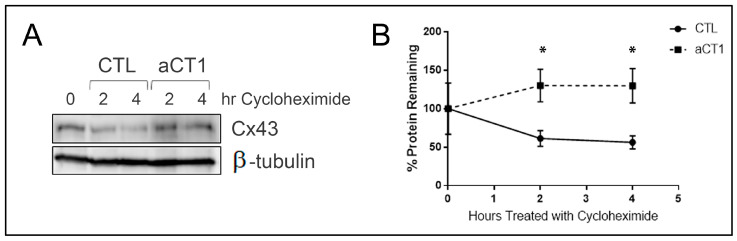
aCT1 stabilizes Cx43 protein. (**A**) Western blot of JIMT-1 cells treated with PBS or 100 μM aCT1 in the presence of cycloheximide. (**B)** Quantitation of Cx43 protein corrected for loading with β-tubulin from n = 3 individual western blots. Student’s *t*-test, unpaired with Welch’s correction * *p* < 0.05. The uncropped blots are shown in Appendix A.

**Figure 3 cancers-16-00423-f003:**
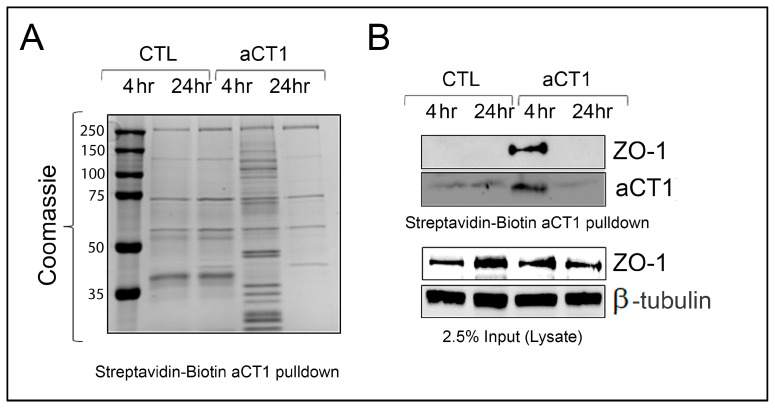
Proteomic analysis of proteins that bind to aCT1. (**A**) Streptavidin matrix was used to pulldown Biotin-tagged aCT1 from JIMT-1 cells treated with peptide for indicated time. Coomassie stained gel showing that after 4 hrs of 100 μM aCT1 treatment, a number of proteins bind to aCT1. (**B**) Western blot analysis showing that aCT1 binds to ZO-1. (Top—pulldown; bottom—input from lysate). The uncropped blots are shown in Appendix A.

**Figure 4 cancers-16-00423-f004:**
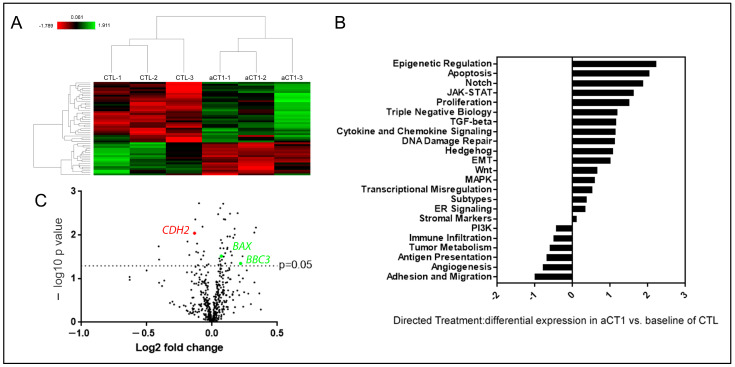
aCT1 impacts gene expression. (**A**) Heat map showing genes that were significantly upregulated (in green) and downregulated (in red) in JIMT-1 cells comparing CTL and aCT1 treatment. (**B**) Directed differential expression analysis of aCT1- vs. CTL- treated JIMT-1 cells and sorted into the pathways represented by the NanoString Breast Cancer 360 nCounter panel. (**C**) Volcano plot of individual genes that were altered in aCT1 vs. CTL treated JIMT-1 cells. Upregulated genes *BAX* and *BBC3* outlined in green and downregulated gene *CDH2* outlined in red. Analysis performed using the nSolver 4.0 software.

**Figure 5 cancers-16-00423-f005:**
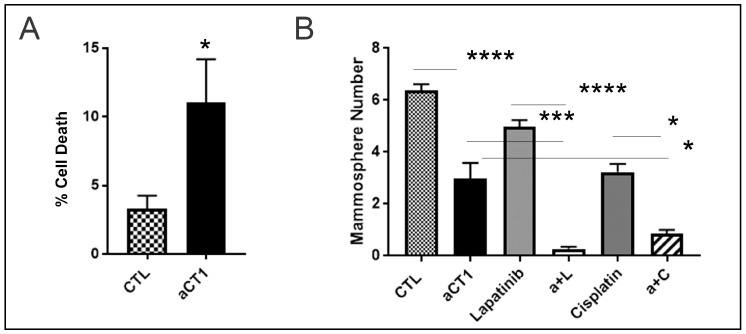
aCT1 promotes cell death. (**A**) JIMT-1 cells treated with PBS as a control (CTL) or 250 μM aCT1 were analyzed in monolayer culture using trypan blue exclusion. Unpaired student’s t-test with Welch’s correction * *p* < 0.05. (**B**) JIMT-1 cells treated with PBS as a CTL, 50 μM aCT1, 100 nM lapatinib, 500 nM of cisplatin, a combination of aCT1 + laptinib (a + L) or aCT1 + cisplatin (a + C) were assessed by mammosphere assay. One-way ANOVA with Tukey’s multiple comparisons. * *p* < 0.05, *** *p* < 0.001, **** *p* < 0.0001.

**Table 1 cancers-16-00423-t001:** Top pathways from pathway analysis examining the proteins significantly bound to aCT1 compared to CTL.

Category	Term Description	Observed	Background	FDR
GO Process	Gene expression (GO:0010467)	111	2101	1.38 × 10^−22^
GO Function	RNA binding (GO:0003723)	107	1672	1.12 × 10^−28^
KEGG	Ribosome (hsa03010)	22	131	6.45 × 10^−12^
Reactome	Metabolism of RNA (HSA-8953854)	58	705	1.46 × 10^−18^

**Table 2 cancers-16-00423-t002:** Pathway analysis of aCT1 altered genes.

Category	Term Description	Observed	Background	FDR
GO Process	Cell cycle process (GO:0022402)	19	976	2.81 × 10^−8^
GO Function	Protein binding (GO:0005515)	39	7026	1.40 × 10^−5^
KEGG	Pathways in cancer (hsa05200)	15	517	1.13 × 10^−9^
Reactome	Cell junction organization (HSA-446728)	7	92	1.41 × 10^−5^

## Data Availability

Raw and processed mass spectrometry data are available through the MassIVE Repository (a ProteomeXchange partner. Accession MSV000093550; Reviewer login information username Reviewer_ MSV000093550; Password: Act1). nCounter data are available at NCBI GEO (GSE250543). Data are also available in Appendix A.

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
