# Peer review of "Mapping the Anti-Cancer Activity of α-Connexin Carboxyl-Terminal (aCT1) Peptide in Resistant HER2+ Breast Cancer"

_cancers, 2024, doi:10.3390/cancers16020423_

Round 1

Reviewer 1 Report

Comments and Suggestions for Authors

The study, entitled "Mapping the Anti-Cancer Activity of α-Connexin Carboxyl-Terminal (aCT1) Peptide in Resistant HER2+ Breast Cancer", seeks to the anti-cancer mechanisms of a therapeutic peptide known as alpha-connexin carboxyl-terminal peptide (aCT1) targeting connexin 43 (Cx43). The study reconfirms that aCT1 enhances intercellular communication at gap junctions and stabilizes the Cx43 protein, which is consistent with previous findings on the mechanism of action of this compound. In addition, the study highlights the interaction of aCT1 with the ZO-1 protein and uncovers novel proteins that interact with aCT1 and its influence on breast cancer signaling pathways.

To increase the clarity and impact of this study, the following improvements are suggested:

Request for a graphical schematic: authors are encouraged to provide a detailed graphical representation of the mechanism of action of the aCT1 peptide and its interactions in the cellular environment. This will help to visually summarize the complex interactions and pathways involved.

Comparison with published data: The authors should make a comprehensive comparison with previously published data, discussing both the similarities and the novelties of their findings. This comparison could help to highlight the unique contributions of this study.

Methodology of proteomic analysis: Detailed documentation of the proteomic analysis methods is essential. Authors should provide a detailed description of the techniques, instruments and protocols used, including all quantification and validation strategies. This level of detail helps with the reproducibility and verification of the methods used.

Commitment to data sharing: Authors are encouraged to deposit their raw data in a recognized public data repository. Access to the raw data underlines the transparency of the research and enables other scientists to perform validation and further analysis. The chosen repository should be reputable and widely used in the industry to ensure accessibility and long-term availability.

Author Response

Please see attachment. Please find responses to your comments under Reviewer 1. Thank you.

Reviewer 2 Report

Comments and Suggestions for Authors

Overall, this is a well-written and highly organized manuscript. The authors present a work of how the aCT1 peptide mediates the gap junction intercellular communication of Cx43 on the resistant HER+ breast cancer cells, and how the downstream biological functions are affected. To validate the mechanisms, cell coupling analysis, proteomics analysis, and genomics analysis were conducted. The shown data are adequate to support the statement. I only have some minor comments as the following:

1.       Please add the full name of ZO-1 in the simple summary as well.

2.       Fix the Typo in line 59 “Cx43-trageted”.

3.       Please rephrase the last sentence of the intro (line 74-76).

4.       Please elaborate how the biotinylation of aCT1 was performed, and what was used to elute the captured proteins. If the aCT1 product is biotinylated, please add the explanation in the text.

5.       Typo in line 258 “with with”

Author Response

Please see attachment. Responses to your review are under Reviewer 2. Thanks.

Round 2

Reviewer 1 Report

Comments and Suggestions for Authors

The authors have performed the reviewer suggestions.